# MulFCoder: Framework-conditioned Multi-agent for MLLM-based Multi-framework Front-end Code Generation

Jie Wu [* 1]  Haoran Ma [* 2]  Shisong Tang [3]  Yulin Xu [4]  Xiaoyu Kang [5]  Jiechao Gao [† 6]

## Abstract

Generating runnable front-end code from UI screenshots is a long-standing goal in automated software engineering. Existing MLLM-based methods predominantly focused on HTML/CSS, leaving multi-framework generation for React/Vue/Angular underexplored. Naively modifying prompts leads to substantial performance gaps across multi-framework and highly framework-specific error modes. To address this, we propose **MulFCoder**, a framework-conditioned multi-agent method that explicitly encodes framework constraints to bring multi-framework differences into a decidable rule space. MulFCoder orchestrates four agents: Grounder constructs an ElementTable, ContentTable, and macro-layout regions from detected UI elements; Planner builds a DOM-like hierarchical layout tree, produces a task schedule, and derives a framework-specific file Contract; Writer generates structured file writes or patches within a restricted edit window; Judger enforces lightweight, framework-conditioned constraints to accept or reject updates and trigger bounded repairs, preventing drift and deadlocks without expensive builds. Experiments demonstrate that MulFCoder substantially improves compilation success rate and reduces framework-specific errors, with particularly pronounced gains on Angular.

## 1. Introduction

Generating runnable front-end code from UI screenshots is a long-standing goal in automated software engineering. It can lower the barrier of implementation and improve development efficiency and delivery quality (Fan et al., 2023). Traditional front-end code generation methods utilize deep learning and computer vision techniques to extract structured information for generating HTML/CSS code (Chen et al., 2024a;c; 2023).

With multimodal large language models (MLLMs), UI-to-code is moving from element extraction pipelines toward directly synthesizing executable code from pixels by combining visual understanding with code generation capability. Yet, straightforward prompting remains unreliable because UI synthesis requires precise grounding and layout and style preservation, leading to omissions, distortions, and misarrangements in generated outputs (Wan et al., 2025). To narrow this gap, researchers scale screenshot–code data and evaluation via large synthetic and real datasets and visual-fidelity benchmarks (Laurençon et al., 2024; Si et al., 2025), and adopt structure- and layout-aware and agentic workflows that decompose planning and generation to better retain hierarchy and alignment (Gui et al., 2025b; Jiang et al., 2025). Emerging work further targets executability and fidelity through post-training (e.g., visual reinforcement learning) (Zhao et al., 2025) and extends beyond page-level HTML toward finer UI components with task-specific representations (Zhang et al., 2025).

However, current MLLM-based methods are mainly focused on HTML/CSS generation, and lack systematic study and unified modeling for multi-framework generation in mainstream engineering frameworks such as React/Vue/Angular. A straightforward idea is to switch the generation target to different frameworks only by modifying the prompt (Li et al., 2025), but such "prompt transfer" often leads to substantial performance drops: compilation success varies greatly across frameworks, accompanied by highly framework-specific error modes. (Sendyka et al., 2025) The root cause is that multi-framework differences are not merely syntactic sugar replacement, but joint changes in language rules, component boundary conventions, and project scaffolding constraints (Xing et al., 2025); without explicitly modeling these constraints, the model is prone to drift during long-horizon generation, and errors are difficult to localize and fix reliably. Ultimately, this results in many invalid attempts and manual rework in real production, off-

---

[*]Equal contribution [†]Corresponding author. [1]Communication University of China [2]Harvard University [3]Tsinghua university [4]University of Southern California [5]Chinese Academy of Sciences [6]Stanford University. Correspondence to: Jiechao Gao <jiechao@stanford.edu>.

*Proceedings of the 43rd International Conference on Machine Learning*, Seoul, South Korea. PMLR 306, 2026. Copyright 2026 by the author(s).

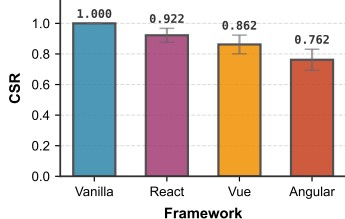

*(a)* Compilation Success Rate.

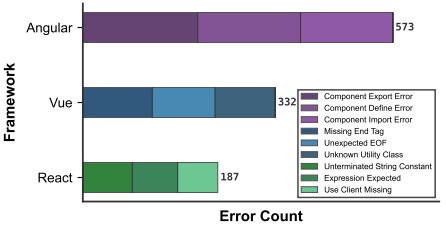

*(b)* Compilation Error Distribution.

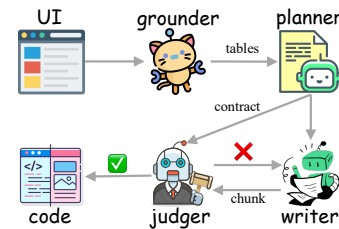

*(c)* Core idea of MulFCoder.

*Figure 1.* (a) On Design2Code dataset, five LLMs that generate code by directly rewriting prompts exhibit a substantial gap in compilation success rate (CSR) across frameworks, indicating that multi-framework discrepancies cannot be addressed by prompt changes alone and are instead driven by framework specific syntax and engineering constraints; (b) under the same setup, the Top-3 compilation error distributions of the five LLMs across three frameworks show clear framework specific error patterns; (c) an overview of the MulFCoder.

setting the efficiency gains brought by automation.

To address these issues, we propose MulFCoder, a framework-conditioned multi-agent method that explicitly encodes framework constraints to "collapse" multi-framework differences into a decidable and executable rule space, thereby improving cross-framework consistency and executability. The core intuition is to decompose UI→Code from one-shot long-text generation into a collaborative process with clear role specialization and stepwise verification, and to drive generation and repair with framework-conditioned hard constraints. Firstly, the Grounder starts from detected UI elements to construct an ElementTable, a ContentTable, and macro-layout regions, so that content binding and the spatial skeleton are fixed early. Secondly, the Planner organizes element relations within each macro region to build a DOM-like hierarchical layout tree, outputs a file-level task schedule, and derives a framework-specific file Contract to restrict the editable window. Finally, the Writer produces structured file writes or patches within the restricted window, while the Judger applies lightweight, framework-conditioned FastGate rules to accept or reject updates and triggers bounded repairs to prevent drift and deadlocks. Because constraints are made explicit and embedded into the collaborative loop, MulFCoder can consistently suppress framework-specific errors and improve multi-framework executability without relying on expensive builds.

To sum up, our contributions are as follows:

- We introduce and systematically study the problem of multi-framework generation for MLLM-based front-end code, and reveal the compilation discontinuities and highly framework-specific error modes induced by prompt-only transfer.

- We propose MulFCoder, a framework-conditioned multi-agent method that uses a modular Grounder, Planner, Writer, and Judger decomposition and a file-level contract to bring framework differences into a

decidable rule space.

- We conduct comprehensive experiments on multi-framework benchmarks, demonstrating substantial improvements in compilation success, and we release the code to facilitate future research.

## 2. Related Work

**Front-end Code Generation Method** Early front-end code generation used end-to-end CNN–RNN image-to-sequence models that mapped screenshots to DSLs (Beltramelli, 2018; Xu et al., 2021). Later work moved to direct HTML generation (Chen et al., 2018) and improved fidelity with explicit layout recognition (Cizotto et al., 2023) or CV pipelines (OCR/object detection) that recover structured UI models (Nguyen & Csallner, 2015; Wu et al., 2021). With MLLMs, progress is driven by large-scale synthesis (Laurençon et al., 2024) and layout-aware, hierarchical planning (Gui et al., 2025c;b), alongside decomposition (Wan et al., 2025; Wu et al., 2025) and efficiency/post-training techniques (Xiao et al., 2025b; Zhao et al., 2025; Chen et al., 2025).

**Front-end Code Generation Benchmark** Benchmarks evolved from screenshot-to-DSL tasks (Beltramelli, 2018; Xu et al., 2021) and OCR/detection-based reconstruction (Nguyen & Csallner, 2015) toward normalized HTML/CSS datasets with reproducible rendering (Lee et al., 2023; Laurençon et al., 2024; Yun et al., 2024; Si et al., 2025; Gui et al., 2025a). Evaluation has shifted from render similarity to visual, element-level metrics (Si et al., 2025; Zhang et al., 2025), including widget/component suites where code alignment is unreliable (Zhou et al., 2025; Soselia et al.). Recent benchmarks add compilation/execution tests and agentic evaluation (Lu et al., 2025; Xiao et al., 2025a; Yang et al., 2025) and scale realism via synthesis and interactive generation (Ge et al., 2025; Wan et al., 2025).

**MLLM**   Multimodal large language models extend pretrained LLMs to images by combining a vision encoder with a language backbone through lightweight connectors that map visual features into token space (Alayrac et al., 2022; Li et al., 2023). Training often aligns modalities with caption/interleaved supervision, then applies visual instruction tuning (Liu et al., 2023; Chen et al., 2024b). Recent models further improve perception and reasoning via higher-resolution tokenization, longer context, and stronger post-training recipes (Liu et al., 2024; Team et al., 2024; Zhu et al., 2025).

## 3. Method

**Overview**   Given a screenshot and a target framework $f = \{\text{React}, \text{Vue}, \text{Angular}, \text{Vanilla}\}$, we aim to produce framework-specific front-end code that is structurally faithful to the UI and, more importantly, satisfies framework-dependent language and engineering constraints. Empirically, different frameworks exhibit markedly different compilation success rates and characteristic failure modes. This indicates that multi-framework generation is not a mere prompt switch, but a change of the admissible program space governed by distinct rules. To explicitly model such differences while keeping inference lightweight, MulFCoder decompose the pipeline into four agents: a *Grounder* that extracts and stabilizes UI elements and contents from the screenshot, a *Planner* that constructs a DOM-like layout tree and simultaneously produces a framework-aware contract that restricts the output space, a *Writer* that generates code incrementally conditioned on the plan and contract, and a *Judger* that applies fast, framework-specific hard checks to accept, reject, or trigger bounded repairs.

### 3.1. Grounder

The Grounder agent converts a screenshot into three structured tables. Here, the element table $\mathcal{E}$ records each UI element's semantic type, BBox, and a stable identifier; the content table $\mathcal{C}$ maintains fill-in contents such as texts and visual references; and the macro-layout $\mathcal{L}$ provides a coarse page partition (e.g., header/sidebar/main/footer) to improve the robustness of downstream agent.

Grounder first applies the UI Element Detection (UIED) algorithm to obtain a set of candidate elements from the screenshot: $\mathcal{E}(x) = \{(\hat{t}_k, \hat{b}_k)\}_{k=1}^{M}$, where $\hat{t}_k$ denotes the element type (e.g., text/button/input/image/icon) and $\hat{b}_k \in [0,1]^4$ is the normalized bounding box. The detector covers common UI primitives such as text blocks, buttons, input fields, images, and icons. These detections serve as the initial input for content binding and structural stabilization.

Text is a critical component of UI semantics, yet detection outputs are often fragmented. To avoid introducing addi-

tional uncertainty in later stages, the Grounder performs OCR and binds recognized text fragments to detected textual elements within this stage. Specifically, we model OCR outputs as $\mathcal{O}(x) = \{(r_j, s_j)\}_{j=1}^{T}$, where $r_j \in [0,1]^4$ is the bounding box of an OCR fragment and $s_j$ is the corresponding text string. We assign each fragment to the most compatible detected element using geometric coverage:

$$\text{assign}(j) = \arg\max_{k} \text{cov}(r_j, \hat{b}_k), \text{cov}(r, b) = \frac{\text{area}(r \cap b)}{\text{area}(r)}. \tag{1}$$

Fragments with insufficient coverage are left unassigned to reduce mismatches. When multiple fragments are assigned to the same element, we aggregate them in reading order, thereby closing the element-to-content association within the Grounder and providing consistent content pointers for subsequent modules.

To ensure consistent references across stages, we assign a stable identifier *eid* to each element. We then construct the content table $\mathcal{C}$: for textual elements, $\mathcal{C}$ stores normalized text along with confidence scores; for visual elements, $\mathcal{C}$ stores visual references (e.g., crop coordinates) that can be used for later fill-in or linking without additional matching. The element table $\mathcal{E}$ and the content table $\mathcal{C}$ are connected via *eid* in a one-to-one or one-to-many manner, forming a unified interface shared by subsequent agents.

Finally, since complex UIs are often organized by a small number of macro regions, the Grounder infers a macro-layout $L$ to coarsely partition the page and assign elements to their corresponding regions. This macro scaffold reduces the influence of local noise on global structure and provides a stable prior for downstream planning and generation.

### 3.2. Planner

Planner takes Grounder outputs, namely the element table $\mathcal{E}$ and the macro layout $L$, together with the target framework $f$, and produces the layout plan and task schedule $P$ as well as a framework contract $K$. Beyond structural organization, Planner also fixes the expected engineering form of each subtask under $f$, turning multi-framework discrepancies into explicit, executable constraints and an admissible output space for downstream generation.

**DOM-like Layout Tree**   Planner first constructs a rooted ordered tree $T$, where internal nodes represent container units (e.g., regions, cards, list blocks), and leaf nodes correspond to grounded element identifiers *eid*. The construction proceeds from the macro layout: for each macro region, Planner creates a region container node and attaches all elements falling within that region as children in reading order. To make layout intent explicit for downstream synthesis, each container node is further annotated with a layout primitive (e.g., row-wise, column-wise, grid, or overlay), which

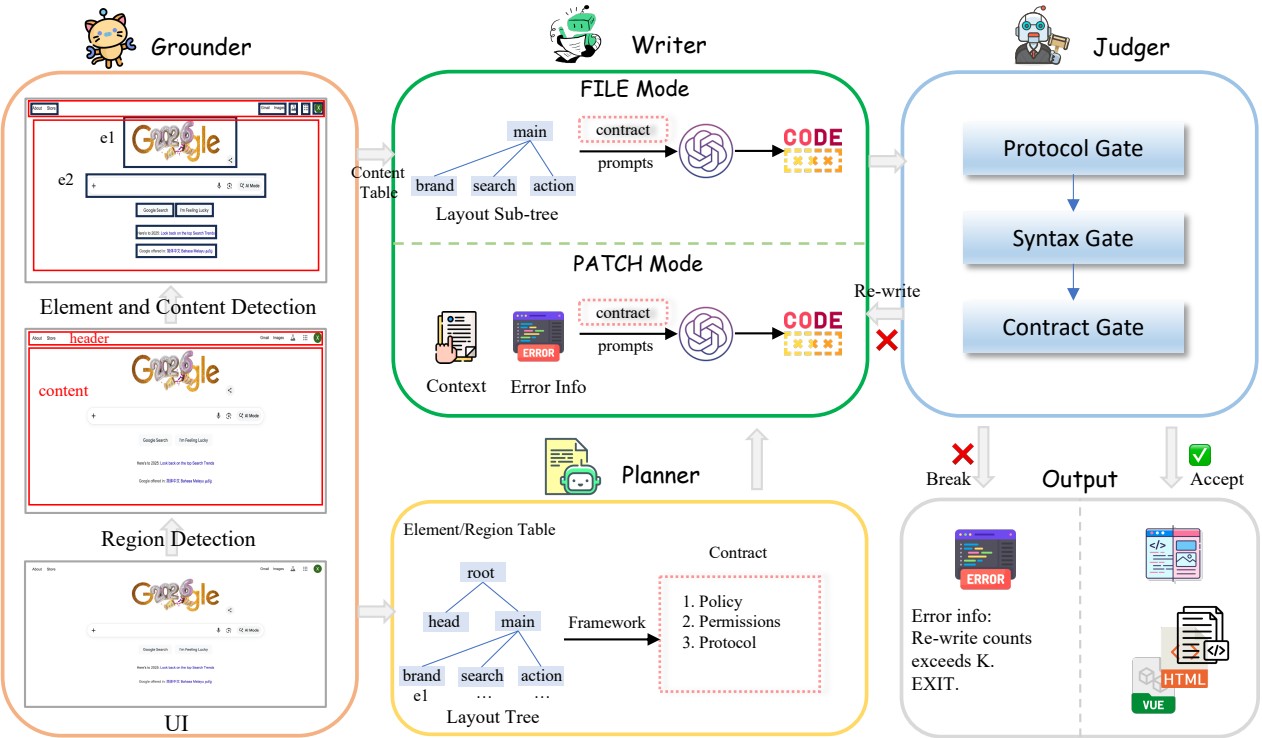

*Figure 2.* Overview of MulFCoder.

provides Writer with a translatable and framework-agnostic structural semantics.

The selection of layout primitives is driven by robust and lightweight geometric heuristics. Intuitively, when child elements are primarily arranged along the horizontal axis, a row-wise primitive is preferred; when the dominant variation is vertical, a column-wise primitive is selected. If a clear two-dimensional alignment pattern emerges, a grid primitive is used. If strong overlap relationships exist (e.g., badges or text-on-image), an overlay primitive is chosen. This design yields a consistent structural semantic input across frameworks, reducing drift caused by Writer re-inventing layout strategies during framework transfer.

**Task Decomposition and Scheduling**   To mitigate the risk of generating an entire page in a single pass and to localize errors for subsequent refinement, Planner decomposes the layout tree $T$ into a task sequence $S = \{\tau_1, \tau_2, ..., \tau_n\}$. Each task is anchored at a subtree root and carries a structural description of the subtree together with an index to the corresponding content slice. The schedule follows a "local-first, assemble-later" principle: it first synthesizes implementations for macro regions or regional subtrees, and then generates the root-level global assembly. This design ensures that local failures do not invalidate the global scaffold and repairs operate within a well-defined scope.

**Contract Construction**   The contract $K$ turns framework differences from implicit prompt into explicit constraints, restricting generation to a framework-consistent feasible set. We decompose $K$ into three parts: framework policy $K_{\text{policy}}$, permission window $K_{\text{perm}}$, and task binding $K_{\text{bind}}$.

The framework policy $K_{\text{policy}}$ encodes framework-level paradigm invariants to prevent cross-framework style mixing. For Angular, it enforces the standalone paradigm by requiring `standalone: true` in `@Component` metadata and explicitly exposing template-scope dependencies via `imports:[*]`, avoiding NgModule-style drift and implicit dependencies. Analogously, for Vue it standardizes the single-file component form while preserving key blocks; for React it enforces TSX functional components with explicit exports; for Vanilla it constrains outputs to HTML structure with modular JS/TS scripts, avoiding framework-specific component syntax.

The permission window $K_{\text{perm}}$ specifies which files are writable and which sensitive regions (e.g., lockfiles and critical configs) are forbidden, preventing the Writer from bypassing constraints by modifying non-target parts and keeping edits within a controlled boundary.

The task binding $K_{\text{bind}}$ constrains the task-level output form. It defines a mapping from a task $\tau$ to a set of framework-specific target forms $\mathcal{T}_\tau^{(f)}$, specifying which engineering units a subtree task should materialize into under framework

$f$ and how the boundaries and dependencies among those units should be expressed. Since engineering forms differ, the same subtree task maps to different legitimate units. For example, the same regional subtree task may correspond to a single SFC file in Vue, a TSX component file in React, a pair of `component.ts` and `component.html` in Angular, and an HTML snippet plus a modular rendering function in Vanilla. The key point here is not "Angular must be standalone," but rather "which concrete component unit(s) this task should generate, how many artifacts it should be split into, and how the root component should reference and assemble them."

### 3.3. Writer

To balance generation quality and multi-framework controllability, Writer agent operates in two modes: FILE mode incrementally generates codes in a task-wise manner, while PATCH mode performs local fixes guided by framework-specific error signals. Both modes share the same input semantics but optimize for different objectives: the former emphasizes controllable generation, whereas the latter targets minimal necessary repair.

**FILE Mode** For task $\tau$, Writer takes subtree $T[\tau]$, local content slice $\mathcal{C}_\tau$, and contract $K$ as inputs. Writer uses $K$ to restrict output space to a feasible region and adopts task-level incremental generation to mitigate long-context drift. During generation, Writer follows a structure-first, content-backfilling order: it first implements the containers and layout skeleton, then instantiates leaf nodes into basic UI primitives, and finally backfills texts and resource references. As a result, even when OCR is noisy or some contents are missing, the structural skeleton is less likely to be disrupted, improving multi-framework consistency. This strategy crystallizes framework differences into $\mathcal{T}_\tau^{(f)}$ and $K$, so that Writer does not need to re-infer engineering conventions for each subtask, which substantially reduces failures in strongly constrained frameworks such as Angular.

**PATCH Mode** When Judger reports errors, Writer switches to PATCH mode and performs minimal necessary repairs within the scope permitted by $K$, bringing the output back into the feasible region. The inputs to PATCH mode consist of four parts: the current candidate artifacts $C$ (restricted to the visible subset within the permission window), the failure feedback (including the failure cluster, localization signals, and evidence), the current subtask $\tau$, and $K$. PATCH mode prioritizes preserving structures that have already passed checks, applies localized fixes around the constraint-triggering points, and avoids introducing new structural rearrangements whenever possible.

### 3.4. Judger

Judger agent imposes lightweight gated checks on generated outputs without invoking full compilation. Its goal is to translate multi-framework differences into decidable necessary conditions and attribute failures to interpretable error clusters, thereby providing signals for Writer to repair.

**Gated Checks** The judging procedure is organized as a unified gated pipeline that progressively tightens the feasible region from output executability, to framework-level feasibility, and finally to contract consistency. First, Judger verifies that Writer output has an executable structured form, i.e., it can be stably interpreted as a set of write or edit operations and satisfies the required fields and type constraints. This step addresses only system-level usability, ensuring that subsequent checks and repairs have a well-defined target and scope. Second, Judger examines basic parse feasibility under the target framework $f$, filtering artifacts that would fail at the earliest parsing stage, such as obvious bracket imbalance, missing critical template blocks, or unclosed template tags. Third, it checks post-parse dependency and scope consistency, focusing on common engineering constraints that most strongly affect compilation success in multi-framework settings, including the completeness of required imports and declarations, the visibility scope of template references, and whether key metadata is sufficient to establish the linkage between templates and logic. Finally, Judger enforces consistency with contract $K$, requiring that generation touches only contract-allowed targets, avoids protected regions, and ensures that contract-mandated key artifacts exist and match the intended task-level output form. Through these gated checks, Judger can, at low cost, reliably compress outputs toward a framework-consistent feasible region and provide clear attribution and localization signals for subsequent PATCH repairs.

**Deadlock Avoidance** Since checking and repair are performed iteratively, the system may repeatedly attempt fixes for the same failure pattern and fall into a loop. To prevent deadlocks, we allocate a bounded repair budget for each subtask and signature the failure feedback to detect substantive progress: after Writer applies a PATCH, Judger re-evaluates and returns updated feedback; if the failure pattern remains unchanged across multiple iterations, the system terminates the branch and rolls back to the most recent state.

## 4. Experiments

### 4.1. Setup

We evaluate on two public benchmarks, Design2Code (Si et al., 2025) and DesignBench (Xiao et al., 2025a). We test MulFCoder on five representative multimodal LLM backbones, namely Gemini-2.5-Pro, GPT-4o, Doubao-Seed-1.6,

*Table 1.* Performance comparison of MulFCoder across different frameworks. In the Vanilla setting, we observe consistent visual improvements. However, in strict engineering frameworks (React, Vue, Angular), we notice an "Engineering Trade-off": while MulFCoder significantly repairs build errors (boosting CSR), this occasionally necessitates simplifying complex UI components, leading to saturated or slightly regressed visual similarity scores (CLIP).

| Base Model | Design2Code | | DesignBench | | Design2Code | | DesignBench | |
|---|---|---|---|---|---|---|---|---|
| | CLIP | CSR | CLIP | CSR | CLIP | CSR | CLIP | CSR |
| **VANILLA** | | | | | **REACT** | | | |
| Gemini-2.5-Pro | 0.92 | 1.00 | 0.83 | 1.00 | 0.91 | 0.97 | 0.84 | 0.91 |
| + *MulFCoder* | **0.94**$^{+0.02}$ | 1.00 | **0.87**$^{+0.04}$ | 1.00 | **0.93**$^{+0.02}$ | **0.98**$^{+0.01}$ | **0.85**$^{+0.01}$ | **0.96**$^{+0.05}$ |
| GPT-4o | 0.91 | 1.00 | 0.81 | 1.00 | 0.89 | 0.96 | 0.79 | 0.94 |
| + *MulFCoder* | **0.93**$^{+0.02}$ | 1.00 | **0.84**$^{+0.03}$ | 1.00 | **0.90**$^{+0.01}$ | **0.98**$^{+0.02}$ | **0.81**$^{+0.02}$ | **0.98**$^{+0.04}$ |
| Doubao-Seed-1.6 | 0.89 | 1.00 | 0.77 | 1.00 | 0.85 | 0.94 | 0.75 | 0.88 |
| + *MulFCoder* | **0.92**$^{+0.03}$ | 1.00 | **0.81**$^{+0.04}$ | 1.00 | **0.87**$^{+0.02}$ | **0.97**$^{+0.03}$ | **0.76**$^{+0.01}$ | **0.93**$^{+0.05}$ |
| Qwen2.5-VL-7B | 0.88 | 1.00 | 0.71 | 1.00 | 0.84 | 0.89 | 0.72 | 0.80 |
| + *MulFCoder* | **0.91**$^{+0.03}$ | 1.00 | **0.76**$^{+0.05}$ | 1.00 | **0.86**$^{+0.02}$ | **0.95**$^{+0.06}$ | **0.75**$^{+0.03}$ | **0.89**$^{+0.09}$ |
| LLaVA-v1.5-7B | 0.81 | 1.00 | 0.66 | 1.00 | 0.77 | 0.85 | 0.67 | 0.77 |
| + *MulFCoder* | **0.86**$^{+0.05}$ | 1.00 | **0.74**$^{+0.08}$ | 1.00 | **0.81**$^{+0.04}$ | **0.92**$^{+0.07}$ | **0.70**$^{+0.03}$ | **0.85**$^{+0.08}$ |
| **VUE** | | | | | **ANGULAR** | | | |
| Gemini-2.5-Pro | 0.85 | 0.93 | 0.79 | 0.85 | 0.83 | 0.85 | 0.75 | 0.78 |
| + *MulFCoder* | **0.87**$^{+0.02}$ | **0.97**$^{+0.04}$ | **0.82**$^{+0.03}$ | **0.92**$^{+0.07}$ | **0.85**$^{+0.02}$ | **0.91**$^{+0.06}$ | **0.77**$^{+0.02}$ | **0.84**$^{+0.06}$ |
| GPT-4o | 0.87 | 0.91 | 0.76 | 0.87 | 0.81 | 0.81 | 0.76 | 0.73 |
| + *MulFCoder* | **0.89**$^{+0.02}$ | **0.96**$^{+0.05}$ | **0.78**$^{+0.02}$ | **0.93**$^{+0.06}$ | **0.83**$^{+0.02}$ | **0.90**$^{+0.09}$ | **0.76**$^{+0.00}$ | **0.76**$^{+0.03}$ |
| Doubao-Seed-1.6 | 0.82 | 0.88 | 0.70 | 0.77 | 0.75 | 0.77 | 0.68 | 0.71 |
| + *MulFCoder* | **0.85**$^{+0.03}$ | **0.94**$^{+0.06}$ | **0.72**$^{+0.02}$ | **0.85**$^{+0.08}$ | **0.78**$^{+0.03}$ | **0.88**$^{+0.11}$ | **0.69**$^{+0.01}$ | **0.81**$^{+0.10}$ |
| Qwen2.5-VL-7B | 0.75 | 0.83 | 0.63 | 0.71 | 0.67 | 0.73 | 0.55 | 0.62 |
| + *MulFCoder* | **0.79**$^{+0.04}$ | **0.91**$^{+0.08}$ | **0.65**$^{+0.02}$ | **0.80**$^{+0.09}$ | **0.70**$^{+0.03}$ | **0.85**$^{+0.12}$ | **0.58**$^{+0.03}$ | **0.75**$^{+0.13}$ |
| LLaVA-v1.5-7B | 0.71 | 0.76 | 0.57 | 0.64 | 0.56 | 0.65 | 0.42 | 0.57 |
| + *MulFCoder* | **0.75**$^{+0.04}$ | **0.86**$^{+0.10}$ | **0.60**$^{+0.03}$ | **0.75**$^{+0.11}$ | **0.60**$^{+0.04}$ | **0.78**$^{+0.13}$ | **0.44**$^{+0.02}$ | **0.70**$^{+0.13}$ |

Qwen2.5-VL-7B, and LLaVA-v1.5-7B. We use two complementary metrics to evaluate our method: Compilation Success Rate (CSR) (Xiao et al., 2025a) and CLIP (Radford et al., 2021). More details are provided in Appendix A.

### 4.2. Main Results

Overall, the results in Table 1 corroborate the alignment between the phenomenon we highlight and our method design: without changing the backbone models, MulFCoder systematically improves compilation success rates in multi framework settings, with larger gains on frameworks with stricter engineering constraints.

On CSR, the improvements exhibit a clear framework gradient. For React, where direct prompting already yields a relatively strong baseline, MulFCoder provides moderate yet consistent gains, averaging about 3.8% on Design2Code and about 6.2% on the more challenging DesignBench. This indicates that cross framework gaps are not merely reflected in the number of terminal error messages, but in whether the structured engineering requirements for successful compilation are reliably satisfied. MulFCoder's planning and constraint mechanisms therefore reduce failures caused by missing structural elements in engineering oriented frameworks such as React.

For Vue, the gains are larger, averaging about 6.6% on Design2Code and about 8.2% on DesignBench. This is consistent with Vue's higher sensitivity to template syntax and structural well formedness, where even minor template level inconsistencies can lead to immediate compilation failure. MulFCoder can be viewed as shifting generation from one shot free form writing to structure verifiable generation, thereby substantially reducing these high sensitivity errors.

Angular most directly reflects our core motivation and most clearly illustrates that multi framework gaps are not resolved by prompt changes. Relative to the baseline, MulFCoder improves CSR by about 10.2% on Design2Code and about 9.0% on DesignBench, the largest gains among the three engineering frameworks. This suggests that Angular failures are more concentrated in the engineering constraint chain, including module organization and dependencies, as well

*Table 2.* Compile error fix ratio of MulFCoder

| Model | Design2Code | | | DesignBench | | |
|---|---|---|---|---|---|---|
| | React | Vue | Angular | React | Vue | Angular |
| Gemini. | 35.7% | 57.6% | 40.3% | 55.6% | 47.1% | 27.8% |
| GPT. | 52.6% | 55.8% | 47.3% | 66.7% | 46.7% | 23.6% |
| Doubao. | 51.7% | 50.0% | 47.7% | 46.2% | 37.0% | 37.5% |
| Qwen. | 54.7% | 47.6% | 44.6% | 47.6% | 32.4% | 35.5% |
| LLaVA. | 47.2% | 42.2% | 37.3% | 36.0% | 31.0% | 31.4% |
| Avg. | 48.4% | 50.6% | 43.4% | 50.4% | 38.8% | 31.2% |

as component boundaries and binding rules. MulFCoder's advantage lies in making such constraints explicit and dominant during generation, preventing the model from over focusing on local UI details while missing the engineering scaffold required for compilation.

Across backbones, the relative gains exhibit a strong capability stratification effect: weaker backbones benefit more. This implies that MulFCoder does not rely on the backbone implicitly learning framework rules; instead, it externalizes part of the framework knowledge into executable constraints via structured planning, yielding a stronger shortfall compensation effect when the backbone is underpowered.

For visual similarity, we observe the engineering trade off discussed in the paper. Since CSR is saturated for Vanilla, MulFCoder's contribution is mainly reflected in more consistent visual improvements, with an average relative gain of roughly 3% to 7%, more pronounced on DesignBench. Under strict engineering frameworks such as React, Vue, and Angular, CLIP improvements are closer to a stable non degradation pattern with occasional small gains. This aligns with our design goal: when framework constraints become the primary bottleneck, MulFCoder prioritizes compilation through structural repair and constraint satisfaction, and may adopt more conservative implementations when necessary, resulting in less pronounced visual gains than in Vanilla.

In summary, the main results in Table 1 support our motivation and method choice at two levels. At the macro level, MulFCoder substantially improves CSR in multi framework settings, with gains that closely track the strength of framework engineering constraints, and Angular benefits the most. At the micro level, gains are larger for weaker backbones, suggesting that structured constraints and planning can transform multi framework discrepancies from unconstrained prompt trial and error into a constraint satisfaction problem that can be systematically reduced.

### 4.3. Error Analysis

**Error Fix Rate**    Table 2 summarizes how often MulF-Coder can repair initially non-compiling outputs. Overall, the fix ratio is consistently higher for React and Vue than for Angular, and it further decreases on the more challenging DesignBench. On average, React and Vue stay mostly in the yellow to green range, while Angular drops from the yellow band on Design2Code to predominantly red on DesignBench. This pattern suggests that many React and Vue failures are relatively local and can be resolved through bounded rewriting, whereas Angular failures more often stem from framework-specific engineering constraints, such as dependency wiring and module-level consistency, which are harder to repair without stronger global coordination.

**Error distribution**    Figure 3 characterizes how compilation errors vary across frameworks and how MulFCoder changes the failure landscape. Across two datasets, the distributions are strongly framework-specific rather than model-agnostic: React failures are dominated by JSX parsing and framework specific client boundary constraints, Vue failures concentrate on template well-formedness and attribute mismatches, while Angular failures are primarily driven by project scale engineering constraints such as component wiring, imports, and binding semantics. This observation aligns with our motivation.

Looking into each framework, React errors in Figure 3(a,d) are largely composed of syntax and parsing related categories such as unexpected tokens, unexpected EOF, and expression expected, together with Next.js style constraints such as missing use client. After applying MulFCoder, these dominant categories shrink consistently across all backbones, indicating that enforcing a contract guided plan and validating outputs with a lightweight judge effectively prevents the generation from drifting into malformed TSX or violating client component requirements. For Vue in Figure 3(b,e), the failures are mostly template structure errors, including missing or invalid end tags and related parsing issues, along with style utility mismatches. MulFCoder reduces these high frequency template errors across models, suggesting that the structured planning stage provides a more stable scaffold for template closure and attribute placement, which are otherwise brittle under one shot prompting. For Angular in Figure 3(c,f), the distribution is qualitatively different: errors are concentrated around component import and export, incomplete blocks, binding errors, and other framework level consistency issues. MulFCoder still yields clear reductions in several major categories, but the remaining errors are more persistent and concentrated, reflecting the fact that Angular compilation is especially sensitive to global project coherence across files and metadata, which is harder to satisfy through local edits alone.

Overall, Figure 3 shows two trends: (1) errors exhibit clear framework-specific modes. (2) MulFCoder reduces not only the total number of erros but also the dominance of the most frequent error categories in each framework. The residual gap in Angular further highlights the importance of explicit engineering constraint modeling, and motivates targeted improvements for global consistency in future iterations.

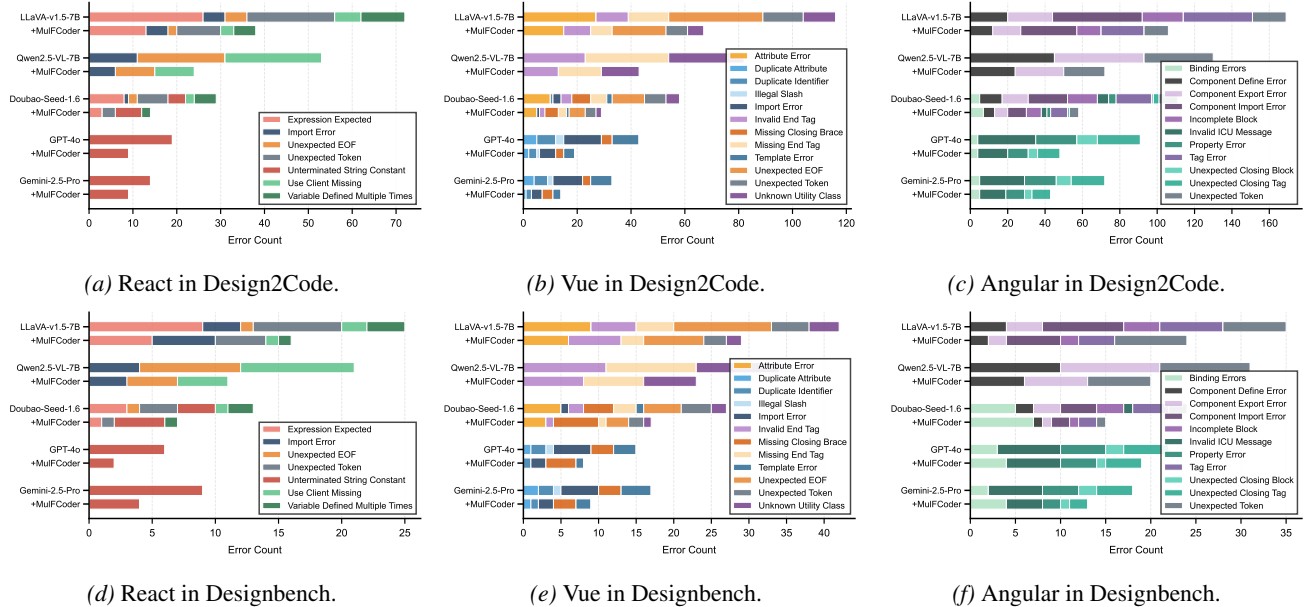

*(a)* React in Design2Code.   *(b)* Vue in Design2Code.   *(c)* Angular in Design2Code.

*(d)* React in Designbench.   *(e)* Vue in Designbench.   *(f)* Angular in Designbench.

*Figure 3.* Compilation error distribution.

## 4.4. Case Study

An example in Figure 4 demonstrates the advantage of MulF-Coder in generation. MulFCoder accurately reproduces structural details, indicating that the fine-grained local code writing achieved through the cooperative multi-agent model significantly improves structural accuracy.

## Conclusion

In this paper, we tackle a critical challenge in MLLM-based UI-to-code generation: producing executable front-end code consistently across mainstream engineering frameworks, where prompt-only transfer induces compilation disconti-nuities and highly framework-specific error modes. Our solution, MulFCoder, is a framework-conditioned multi-agent method that decomposes generation into Grounder, Planner, Writer, and Judger roles and enforces a file-level contract with lightweight, framework-conditioned FastGate verification to keep generation within a decidable and ex-ecutable rule space. Our experiments on multi-framework benchmarks show that this design substantially improves compilation success across React, Vue, and Angular, while also yielding consistent visual improvements in the Vanilla setting. At the same time, we observe an engineering trade-off in strict frameworks: reducing build errors can require simplifying complex UI components, leading to saturated or slightly regressed visual similarity scores. These results suggest that explicitly modeling framework constraints is essential for reliable multi-framework UI synthesis, and we release our code to facilitate future research in this direction.

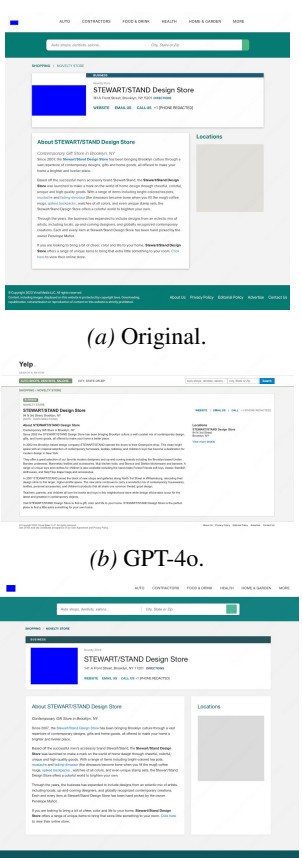

*(a)* Original.

*(b)* GPT-4o.

*(c)* MulFCoder.

*Figure 4.* An UI screenshot generation example.

## Impact Statement

MulFCoder improves multi-framework UI-to-code generation by explicitly modeling framework constraints and using a multi-agent generate-and-repair loop with file-level contracts and lightweight rule-based gating, increasing compilation success across stacks such as React/Vue/Angular. This can reduce repetitive debugging and manual rework for front-end engineers and teams that maintain the same UI across different frameworks, and it can support more reliable, framework-aware evaluation of UI-to-code systems.

Potential risks mirror those of UI code generation systems. Sensitive information in UI artifacts (screenshots, detected elements, extracted content tables) could be exposed if artifacts are logged, retained, or sent to external services without strong governance. Improved front-end generation could also be misused to create deceptive interfaces (e.g., phishing-like pages). Finally, compilation success does not guarantee secure, accessible, or maintainable code; downstream deployment should still rely on standard review, testing, dependency auditing, and security scanning.

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

# A. Setup

## A.1. Dataset

We evaluate our method on two public benchmark datasets, Design2Code (Si et al., 2025) and DesignBench (Xiao et al., 2025a). Design2Code targets the end to end task of generating a web implementation from a UI image. Its data are built from real webpages: the authors crawl webpages linked from the C4 validation split, then apply automatic filtering and manual curation to obtain 484 high quality and diverse test webpages. Each sample provides a webpage screenshot as the visual input, together with a normalized representation of the webpage implementation, such as inlining CSS into HTML to form a self contained artifact, removing external dependencies, and replacing multimedia content with placeholders to reduce uncontrolled variation across environments. DesignBench further extends evaluation to a multi-framework and multi-task benchmark, covering vanilla HTML CSS as well as three mainstream frameworks, React, Vue, and Angular, and evaluating models across generation, editing, and repair stages of the development workflow. It contains 900 webpage samples spanning more than 11 themes, and further categorizes 9 editing types and 6 issue types, enabling systematic analysis of model behavior under different frameworks and task conditions.

## A.2. Baseline

We evaluate MulFCoder in a backbone agnostic manner by instantiating it on top of five representative multimodal LLMs, namely Gemini-2.5-Pro, GPT-4o, Doubao-Seed-1.6, Qwen2.5-VL-7B, and LLaVA-v1.5-7B, and report both the original backbone performance and the corresponding MulFCoder enhanced results. Our baseline is prompt transfer: given the same UI screenshot, we only modify the textual instruction to request code in a target framework, and ask the model to directly generate a runnable project in one shot. The direct prompting results serve as the backbone reference, while the rows marked with "+ MulFCoder" in Table 1 correspond to applying our method on the same backbone under the same target framework.

## A.3. Metrics

We use two complementary metrics to capture both visual fidelity and executability across frameworks:

**Compilation Success Rate (CSR)** (Xiao et al., 2025a) measures executability. For each generated sample under a target framework, CSR counts it as successful if the produced project can pass the framework compilation pipeline without errors, and reports the success ratio over all samples. We define CSR as the ratio of generated samples that pass the framework-specific build process without fatal errors:

$$\text{CSR}_f = \frac{1}{N} \sum_{i=1}^{N} \mathbb{I}(\text{Build}_f(y_i) = \text{Success}), \tag{2}$$

where $N$ is the total number of test samples, $y_i$ is the generated code for the $i$-th sample, and $\text{Build}_f$ represents the standard build toolchain (e.g., Vite or Webpack) for framework $f$. This metric directly reflects the practical usability of generated code and is the key indicator for cross-framework reliability.

**Visual Similarity (CLIP Score)** (Radford et al., 2021) measures visual similarity between the input UI screenshot and the rendered output of the generated implementation. We compute an image level similarity score using CLIP and report the dataset level average. For samples that successfully compile, we verify their visual correctness. We render the generated code into a screenshot $\hat{x}_i$ using a headless browser and compare it against the original ground truth screenshot $x_i$. We utilize the CLIP score to measure semantic and structural alignment:

$$\text{ScoreCLIP} = \cos(E\text{visual}(x_i), E_{\text{visual}}(\hat{x}_i)), \tag{3}$$

where $E$visual denotes the pre-trained CLIP vision encoder (ViT-B/32). We choose CLIP over pixel-wise metrics (like RMSE) because it correlates better with human perception of layout similarity and is robust to minor rendering artifacts (e.g., slight font weight differences) that do not affect usability.

# B. Error Case Study

**React case study**    Listing 1 and 2 show a representative React failure produced by GPT-4o under direct prompting and the corresponding fix generated by MulFCoder. In the original code, a typical *Expression Expected* compilation error is triggered at a conditional rendering site, where the ternary expression is left incomplete as `post.hasThumbnail ?` `<div ...></div> :`. Since the `:` branch does not provide a valid JSX sub-expression, the parser cannot construct a well-formed AST and fails immediately. Such errors are common in long-form generation, where the model may drift in bracket and branch closure, especially for high-frequency patterns like JSX conditionals. MulFCoder resolves this issue with a minimal, structure-preserving edit by rewriting the fragment into a fully specified conditional expression, `post.hasThumbnail ? ( ... ) : null`. This fix explicitly completes the else branch and stabilizes the syntactic structure via parentheses, while preserving the original UI logic and layout semantics. Overall, this case illustrates that MulFCoder can localize compilation failures to verifiable structural defects and apply constrained, minimal edits to recover compilable React code.

**Vue case study**    As shown in the Listing 3 and 4, the Vue template produced by direct prompting triggers a typical *Attribute Error* at a binding site, whereas MulFCoder restores compilability via a local, structure-preserving edit. Specifically, the original implementation binds `:src="post.thumbnail.url"` directly on the `` element, implicitly assuming that `post.thumbnail` is always present. When this field is null or undefined, template rendering or static checks can raise an invalid property access, leading to compilation failure. This failure mode reflects that one-shot generation often overlooks field nullability and template-level defensive patterns. MulFCoder fixes the issue without changing the DOM hierarchy or styling semantics by adding a guard condition `v-if="post.thumbnail"` on the thumbnail container, ensuring that the subsequent `.url` access is evaluated only when the field exists. Overall, this case demonstrates that MulFCoder can attribute failures to missing template-level constraints and apply minimal conditional guards to improve robustness and compilation success in Vue.

**Angular Case Study.**    Listing 5 and 6 present a representative Angular failure from GPT-4o under direct prompting and the corresponding correction produced by MulFCoder. The original template mixes incompatible binding syntaxes by writing `[style.background-color]=" themeColor ":` it combines Angular property binding brackets with interpolation braces, which violates Angular's template grammar and triggers a binding-related compilation error. This failure mode is characteristic of Angular, where seemingly minor syntax deviations are not tolerated because template parsing and type checking enforce a strict separation between property bindings (which expect an expression) and interpolations (which are used in text nodes and certain attribute contexts). MulFCoder fixes the issue with a minimal, semantics-preserving edit by rewriting the binding as `[style.background-color]="themeColor"`, supplying a valid Angular expression without changing the DOM structure, layout semantics, or surrounding content. Overall, this case highlights that many Angular failures arise from framework-specific template binding rules rather than general HTML syntax, and that MulFCoder can reliably restore compilability by enforcing framework-consistent binding forms through constrained edits.

## B.1. React

*Listing 1.* Wrong React code by GPT4o.

```
1  <div className="popular-post-item mb-12 border-b pb-8">
2    <div className="flex flex-row justify-between">
3      <div className="post-content flex-1 pr-6">
4        <h3 className="text-2xl font-bold text-[#008f76] mb-2 cursor-pointer
                hover:underline">
5          {post.title}
6        </h3>
7        <div className="text-xs font-bold text-gray-500 uppercase mb-3">
8          {post.date}
9        </div>
10       <p className="text-base leading-relaxed text-gray-800 mb-4 font-serif">
11         {post.excerpt}
12       </p>
13     </div>
14
15     {/* Expression Expected */}
16     { post.hasThumbnail ?
17       <div className="w-48 h-48 bg-black shrink-0"></div> :
18     }
19
20   </div>
21
22   <div className="flex justify-between items-center text-[10px] font-bold tracking-widest
          text-[#008f76] uppercase mt-2">
23     <button className="hover:text-[#006f5b]">Share</button>
24     <button className="hover:text-[#006f5b]">Read More</button>
25   </div>
26 </div>
```

*Listing 2.* Right React code by MulFCoder.

```
1  <div className="popular-post-item mb-12 border-b pb-8">
2    <div className="flex flex-row justify-between">
3      <div className="post-content flex-1 pr-6">
4        <h3 className="text-2xl font-bold text-[#008f76] mb-2 cursor-pointer
                hover:underline">
5          {post.title}
6        </h3>
7        <div className="text-xs font-bold text-gray-500 uppercase mb-3">
8          {post.date}
9        </div>
10       <p className="text-base leading-relaxed text-gray-800 mb-4 font-serif">
11         {post.excerpt}
12       </p>
13     </div>
14
15     {/* Fixed */}
16     { post.hasThumbnail ? (
17       <div className="w-48 h-48 bg-black shrink-0"></div>
18     ) : null }
19
20   </div>
21
22   <div className="flex justify-between items-center text-[10px] font-bold tracking-widest
          text-[#008f76] uppercase mt-2">
23     <button className="hover:text-[#006f5b]">Share</button>
24     <button className="hover:text-[#006f5b]">Read More</button>
25   </div>
26 </div>
```

## B.2. Vue

*Listing 3.* Wrong Vue code by GPT4o.

```
1  <template>
2    <div class="popular-posts-section">
3      <h3>Popular posts from this blog</h3>
4
5      <div class="post-list">
6        <div v-for="post in popularPosts" :key="post.id" class="post-item">
7          {/* Attribute Error */}
8          <div class="media-left">
9            
10         </div>
11
12         <div class="content-body">
13           <h4 class="title">{{ post.title }}</h4>
14           {{ post.date }}
15           <p class="excerpt">{{ post.excerpt }}</p>
16           <div class="actions">
17             <a href="#">SHARE</a>
18             <a href="#" class="read-more">READ MORE</a>
19           </div>
20         </div>
21       </div>
22     </div>
23   </div>
24 </template>
```

*Listing 4.* Right Vue code by MulFCoder.

```
1  <template>
2    <div class="popular-posts-section">
3      <h3>Popular posts from this blog</h3>
4
5      <div class="post-list">
6        <div v-for="post in popularPosts" :key="post.id" class="post-item">
7          {/* Fixed */}
8          <div class="media-left" v-if="post.thumbnail">
9            
10         </div>
11
12         <div class="content-body">
13           <h4 class="title">{{ post.title }}</h4>
14           {{ post.date }}
15           <p class="excerpt">{{ post.excerpt }}</p>
16           <div class="actions">
17             <a href="#">SHARE</a>
18             <a href="#" class="read-more">READ MORE</a>
19           </div>
20         </div>
21       </div>
22     </div>
23   </div>
24 </template>
```

### B.3. Angular

*Listing 5.* Wrong Angular code by GPT4o.

```
1  <!-- post-hero.component.html -->
2  <div class="container">
3    <header class="top-nav">
4      DIGITAL INSPIRATION
5      <a href="#" class="search-link">SEARCH</a>
6    </header>
7
8    {/* Binding Errors */}
9    <section class="hero-section" [style.background-color]="{{ themeColor }}">
10     <div class="hero-content">
11       <h1>How to Upload Videos to YouTube if Access to Website is Blocked</h1>
12
13       <div class="post-meta">
14         July 25, 2007
15       </div>
16     </div>
17   </section>
18
19   <article class="post-body">
20     <p class="drop-cap">
21       Richard, who works for a large Fortune 500 company, recently wrote on Facebook...
22     </p>
23     <a href="#" class="share-btn">SHARE</a>
24   </article>
25 </div>
```

*Listing 6.* Right Angular code by MulFCoder.

```
1  <!-- post-hero.component.html -->
2  <div class="container">
3    <header class="top-nav">
4      DIGITAL INSPIRATION
5      <a href="#" class="search-link">SEARCH</a>
6    </header>
7
8    {/* Fixed */}
9    <section class="hero-section" [style.background-color]="themeColor">
10     <div class="hero-content">
11       <h1>How to Upload Videos to YouTube if Access to Website is Blocked</h1>
12
13       <div class="post-meta">
14         July 25, 2007
15       </div>
16     </div>
17   </section>
18
19   <article class="post-body">
20     <p class="drop-cap">
21       Richard, who works for a large Fortune 500 company, recently wrote on Facebook...
22     </p>
23     <a href="#" class="share-btn">SHARE</a>
24   </article>
25 </div>
```

## C. Performance under different difficulty levels.

*Table 3.* CSR of three frameworks under different difficulty levels in DesignBench.

| Model | Easy | | | Medium | | | Hard | | |
|---|---|---|---|---|---|---|---|---|---|
| | React | Vue | Angular | React | Vue | Angular | React | Vue | Angular |
| **Gemini-2.5-Pro+** | 1.00 | 0.98 | 0.95 | 1.00 | 0.86 | 0.85 | 0.89 | 0.89 | 0.62 |
| **GPT-4o+** | 1.00 | 1.00 | 0.87 | 1.00 | 0.86 | 0.75 | 0.93 | 0.89 | 0.44 |
| **Doubao-Seed-1.6+** | 1.00 | 0.96 | 0.93 | 0.89 | 0.86 | 0.80 | 0.88 | 0.66 | 0.57 |
| **Qwen2.5-VL-7B+** | 0.95 | 0.92 | 0.85 | 0.85 | 0.83 | 0.75 | 0.84 | 0.50 | 0.40 |
| **LLaVA-v1.5-7B+** | 0.96 | 0.87 | 0.81 | 0.85 | 0.76 | 0.70 | 0.66 | 0.30 | 0.10 |

