# OpenReview forum: "MulFCoder: Framework-conditioned Multi-agent for MLLM-based Multi-framework Front-end Code Generation"
_ICML.cc/2026/Conference — ICML 2026 regular_

### Official Review · Reviewer_PcxA · 2026-02-23

**Soundness:** 3
**Presentation:** 3
**Significance:** 3
**Originality:** 3
**Overall Recommendation:** 5
**Confidence:** 4

**Summary:**

This paper proposes MulFCoder, a framework-conditioned multi-agent system designed for executable front-end code generation using UI screenshots, applicable across various frameworks such as React, Vue, and Angular. The process of UI-to-code generation is decomposed into four phases, namely, grounder, planner, writer, and judger. The two propositions underlying the framework are as follows: first, the integration of engineering constraints related to frameworks into contract-guided generation; and, second, an iterative evolutionary approach towards code repair, reducing compilation failures. The paper presents significant reductions in framework-related errors, typically encountered during cross-framework code generation using LLM, even with naive rewriting of prompts. The empirical results show improvements in the compilation success rate using Design2Code, DesignBench, and various multimodal LLM backends, while maintaining competitive visual similarity.

**Compliance With Llm Reviewing Policy:**

Affirmed.

**Key Questions For Authors:**

1.The paper focuses on Vanilla/React/Vue/Angular. How do you expect the framework to extend to other stacks?

2.You mention an “engineering trade-off” where enforcing executability may simplify complex UI components and lead to slightly regressed CLIP under strict frameworks. Could you provide a bit more qualitative breakdown of when this happens (e.g., which UI patterns are most affected)?

**Limitations:**

yes

**Strengths And Weaknesses:**

## Soundness

The multi-agent decomposition solution is clearly described. The motivation for splitting long-horizon UI-to-code generation into multiple stages is very reasonable. The experiments cover various backbones and two public benchmarks. The improvements in compilation success rate are consistent across frameworks.

However, the gains in visual similarity are sometimes relatively small compared to compilation success rate. Thus, there exists a trade-off between executability and UI fidelity, which is not deeply analyzed. Also, usability studies or human-judged correctness may further strengthen the experiments.

## Presentation


The structure of this paper is nice. The pipeline is explained very clearly with intuitive modular. The architectural figures and error distribution plots are helpful for understanding the design choices.

However, some aspects of framework contract implementation and judging heuristics is a little bit high-level.


## Significance

The paper aims a practically relevant problem in multimodal code generation and automated software engineering. Improving cross-framework executability is important for real deployment of UI-to-code systems. Thus, the reported improvements have direct practical value.

 Although the authors suggest broader applicability, the experiments are limited to front-end frameworks. It remains unclear how well the approach transfers to other domains.

## Originality

Applying multi-agent decomposition and structured planning to solve cross-framework code generation is novel Explicit formulation of framework-level contracts and constraint-guided verification loop form a coherent and practically motivated system design.

But some architectural components extend existing agent-style pipelines, and the main novelty lies in the framework-conditioning formulation and system integration rather than fundamentally new modeling techniques.

---

> ### Author Rebuttal · Authors · 2026-03-29
>
> Thank you for the thoughtful and constructive review. We summarize the concerns into four main questions and respond below.
>
> **Q1: CSR vs. CLIP Trade-off**
>
> Due to space constraint, please refer to **Reviewer 6r7h Q4**
>
> **Q2: Framework Contract and Judging Heuristics Too High-Level**
>
> We agree that this part can be described more concretely. In our implementation, the framework contract is not an abstract prompt-level description, but an explicit specification of three operational constraints: framework policy, which defines the required artifact form for each framework, such as React TSX components, Vue SFC structure, or Angular standalone component organization; writable scope, which restricts which files or regions can be created or modified by a given task; and task-to-artifact binding, which maps each planned subtree to a concrete target artifact under that framework. The Judger is also more concrete than the current description may suggest. Its checks are organized as lightweight necessary conditions over four levels: output executability, parse feasibility, dependency or scope consistency, and contract consistency. In practice, this includes malformed structured outputs, bracket or tag imbalance, missing required template blocks, incomplete imports or declarations, invalid bindings or references, and violations of contract-allowed targets. We will revise this part to make clear that both the Contract and FastGate are implemented as explicit framework-specific operational constraints, rather than high-level guidance alone.
>
>
> **Q3: Scalability to Other Frameworks**
>
> At the method level, MulFCoder does not depend specifically on React, Vue, or Angular. The framework-agnostic components mainly include screenshot grounding, structured layout extraction, and task decomposition. What must be added for a new ecosystem are the framework-specific parts of the framework-conditioned Contract and FastGate. Therefore, extending the approach to other front-end stacks does not require redesigning the whole system. It mainly requires re-instantiating the target framework’s artifact form, file organization, binding semantics, and lightweight validation rules. For example, in Next.js, much of the component and JSX foundation is shared with React, so the main extensions would concern app or router conventions, server or client boundaries, routing, and project scaffolding. By contrast, extending to Svelte would likely be more involved, since its single-file component structure, reactive semantics, and template constraints differ more substantially from React, Vue, and Angular, and would therefore require new Contract and FastGate rules. In this sense, the scalability of the framework mainly depends on how far a new front-end stack is from the current contract abstraction, rather than on any inherent restriction of the method to the three frameworks studied in the paper.
>
> **Q4: Originality Weakness**
>
> We believe the novelty of this work does not lie in renaming a generic multi-stage framework, but in formulating multi-framework UI-to-code as a control problem over a framework-specific feasible program space. Existing structured generation methods typically assume a single implicit target program space and mainly focus on task decomposition or generation quality. In contrast, our core finding is that differences across front-end frameworks are not merely prompt-style differences, but fundamentally change the executable program space itself, as evidenced by the systematically different error patterns and compilation bottlenecks across frameworks. Based on this, the key contribution of MulFCoder is not simply chaining planning and repair together, but introducing the framework-conditioned Contract as an explicit control mechanism that turns framework rules from soft prompts into operational constraints, including writable scope, target artifact form, and subtree-to-artifact bindings, together with the corresponding FastGate-based lightweight validation and bounded repair loop. In other words, what is genuinely new is not the stage names, but a mechanism that makes cross-framework differences explicit, executable, and controllable within an iterative generation loop. The fact that the gains are largest on Angular, where framework-level engineering constraints are strongest, further supports that the effectiveness comes from this explicit modeling rather than from a generic structured pipeline.
>
>
> We thank the reviewer again. We hope our response will resolve your concerns, and we look forward hearing from you.

---

> > ### Author Rebuttal · Reviewer_PcxA · 2026-03-31
> >
> > Thank you to the authors for their rebuttal, it addressed most of my concerns. Therefore, I will increase my score to support the paper’s acceptance. It's a good paper.

---

> > > ### Author Response · Authors · 2026-04-01
> > >
> > > We sincerely thank you for raising your score, for your careful review of our work, and for taking the time to revisit it. We are delighted that our rebuttal was able to address your concerns.

---

### Official Review · Reviewer_DhzZ · 2026-03-08

**Soundness:** 3
**Presentation:** 3
**Significance:** 3
**Originality:** 2
**Overall Recommendation:** 4
**Confidence:** 3

**Summary:**

This paper studies the problem of generating executable front-end code from UI screenshots across multiple engineering frameworks, including React, Vue, Angular, and Vanilla HTML. The authors observe that most existing multimodal large language model (MLLM) approaches primarily focus on HTML/CSS generation and that simple prompt transfer across frameworks leads to large drops in compilation success rates and framework-specific failure modes. To address this issue, the paper proposes MulFCoder, a framework-conditioned multi-agent architecture that explicitly models framework constraints during generation.

The proposed system decomposes UI-to-code generation into four agents: Grounder, Planner, Writer, and Judger. The Grounder extracts UI elements and content tables from screenshots; the Planner constructs a DOM-like layout tree and generates framework-specific contracts; the Writer incrementally generates or patches code based on the plan; and the Judger applies lightweight rule-based checks to enforce framework constraints and guide repair. Experiments on the Design2Code and DesignBench benchmarks demonstrate improved compilation success rates across frameworks and models.

Overall, the article's major contribution pertains to improving cross-framework executability in multimodal UI-to-code generation by explicitly modeling framework constraints within an agent-based generation and repair pipeline.

**Compliance With Llm Reviewing Policy:**

Affirmed.

**Key Questions For Authors:**

- One question concerns the robustness of the framework contracts. Since the contracts encode framework-specific engineering rules, it would be useful to understand how sensitive the system is to their design and whether incorrect or incomplete contracts significantly affect generation quality.

- Another question is related to scalability. The current experiments focus on React, Vue, and Angular. It would be interesting to know how easily the approach can be extended to other frameworks or modern front-end stacks such as Next.js or Svelte.

- It would also be helpful if the authors could clarify the relative contribution of the repair loop compared to the planning stage. In particular, how much of the improvement in compilation success rate comes from the Judger-driven repair process versus the initial planning constraints?

**Limitations:**

yes

**Strengths And Weaknesses:**

Strengths:

- The paper addresses a relevant and practically meaningful problem in the context of multimodal code generation. While previous work has primarily focused on HTML/CSS generation, real-world front-end development often requires code generation for specific engineering frameworks. The authors clearly demonstrate that naive prompt-based transfer across frameworks leads to significant compilation inconsistencies and framework-specific errors, which provides a strong motivation for the proposed method.

- Another strength of the work lies in its structured generation design. By decomposing the generation process into grounding, planning, generation, and verification stages, the method introduces a clear separation of responsibilities that helps mitigate long-horizon generation errors. In particular, the notion of a framework-specific contract that restricts the editable region and enforces engineering constraints is an interesting design choice that helps reduce invalid code structures.

- The empirical evaluation is also relatively comprehensive. The method is tested across several multimodal backbone models, including Gemini-2.5-Pro, GPT-4o, Doubao-Seed, Qwen-VL, and LLaVA. The results show consistent improvements in compilation success rate across frameworks, with particularly large gains for Angular, which the paper argues has stronger engineering constraints. These results support the authors’ claim that explicitly modeling framework constraints can improve cross-framework reliability.


Weaknesses:

- Despite the practical relevance of the problem, the methodological novelty of the proposed system appears somewhat limited. Many components of the architecture resemble existing structured generation pipelines that combine planning, generation, verification, and repair. While the application to multi-framework UI code generation is interesting, the core algorithmic ideas do not appear fundamentally new.

- Another limitation is the reliance on heuristic rules in several parts of the pipeline. For example, layout inference relies on geometric heuristics, and the Judger module performs rule-based validation of framework constraints. The paper does not provide a detailed analysis of how sensitive the system is to these heuristics or how easily the framework could generalize to additional front-end ecosystems.

- The evaluation also focuses heavily on compilation success rate as the primary metric. While compilation is an important requirement, it does not necessarily guarantee that the generated code faithfully reproduces the UI design or that it follows good engineering practices. Although the paper reports CLIP-based visual similarity scores, the results suggest that improvements in compilation success may sometimes come at the cost of simplified UI implementations.

- Finally, the paper would benefit from stronger comparisons and ablation studies. In particular, it would be helpful to isolate the contributions of the individual components (e.g., the contract mechanism, the Judger module, and the task decomposition strategy). Without such analyses, it is difficult to determine which parts of the system contribute most to the reported improvements.

---

> ### Author Rebuttal · Authors · 2026-03-29
>
> Thank you for the thoughtful and constructive review. We summarize the concerns into four main questions and respond below.
>
> **Q1: Methodological Novelty**
>
> We believe the novelty of this work does not lie in renaming a generic multi-stage framework, but in formulating multi-framework UI-to-code as a control problem over a framework-specific feasible program space. Existing structured generation methods typically assume a single implicit target program space and mainly focus on task decomposition or generation quality. In contrast, our core finding is that differences across front-end frameworks are not merely prompt-style differences, but fundamentally change the executable program space itself, as evidenced by the systematically different error patterns and compilation bottlenecks across frameworks. Based on this, the key contribution of MulFCoder is not simply chaining planning and repair together, but introducing the framework-conditioned Contract as an explicit control mechanism that turns framework rules from soft prompts into operational constraints, including writable scope, target artifact form, and subtree-to-artifact bindings, together with the corresponding FastGate-based lightweight validation and bounded repair loop. In other words, what is genuinely new is not the stage names, but a mechanism that makes cross-framework differences explicit, executable, and controllable within an iterative generation loop. The fact that the gains are largest on Angular, where framework-level engineering constraints are strongest, further supports that the effectiveness comes from this explicit modeling rather than from a generic structured pipeline.
>
> **Q2: Robustness of Heuristic Rules and Framework Contracts**
>
> In our system these heuristics are used to constrain the search space rather than replace generation itself. The geometric heuristics in Grounder are only used to extract macro-layout, element tables, and content tables as stable structural anchors for downstream planning, rather than directly determining the final code. Similarly, the rule-based checks in Judger are not meant to serve as a full verifier, but as lightweight necessary-condition filters for frequent and locally repairable framework-specific invalid states. In other words, their role is to reject obviously infeasible intermediate states early, not to drive the final output through handcrafted rules. The experimental results already suggest that these heuristics are not overfitted to a particular model or framework, since the method consistently improves CSR across different backbones and shows larger gains in frameworks with stronger engineering constraints. We agree that the current paper does not provide a dedicated sensitivity analysis of these heuristics, and this can be strengthened. At the same time, generalizing to a new front-end ecosystem does not require redesigning the whole system, because the framework-agnostic part lies in Grounder’s structural extraction, while the parts that need to be re-instantiated are mainly the framework-specific Contract and FastGate rules related to artifact form, binding rules, and file organization.
>
> **Q3: CLIP and CSR Trade-off**
>
> Due to space constraint, please refer to **Reviewer 6r7h Q4**
>
>
> **Q4: Ablation Study**
>
> Due to space constraint, please refer to **Reviewer 6r7h Q1**
>
>
> **Q5: Scalability to Other Frameworks**
>
> At the method level, MulFCoder does not depend specifically on React, Vue, or Angular. The framework-agnostic components mainly include screenshot grounding, structured layout extraction, and task decomposition. What must be added for a new ecosystem are the framework-specific parts of the framework-conditioned Contract and FastGate. Therefore, extending the approach to other front-end stacks does not require redesigning the whole system. It mainly requires re-instantiating the target framework’s artifact form, file organization, binding semantics, and lightweight validation rules. For example, in Next.js, much of the component and JSX foundation is shared with React, so the main extensions would concern app or router conventions, server or client boundaries, routing, and project scaffolding. By contrast, extending to Svelte would likely be more involved, since its single-file component structure, reactive semantics, and template constraints differ more substantially from React, Vue, and Angular, and would therefore require new Contract and FastGate rules. In this sense, the scalability of the framework mainly depends on how far a new front-end stack is from the current contract abstraction, rather than on any inherent restriction of the method to the three frameworks studied in the paper.
>
> We thank the reviewer again. We hope our response will resolve your concerns, and we look forward hearing from you.

---

> > ### Author Rebuttal · Reviewer_DhzZ · 2026-04-04
> >
> > My comment has been addressed and I will keep my positive score.

---

> > > ### Author Response · Authors · 2026-04-05
> > >
> > > We sincerely thank you for your careful review of our work. We are delighted that our rebuttal was able to address your concerns.

---

### Official Review · Reviewer_K8yh · 2026-03-08

**Soundness:** 3
**Presentation:** 4
**Significance:** 2
**Originality:** 2
**Overall Recommendation:** 4
**Confidence:** 5

**Summary:**

The paper presents MulFCoder, an approach for front-end code synthesis across multiple frameworks from UI screenshots via a framework-conditioned multi-agent architecture. Specifically, the coordinated Grounder, Planner, Writer, and Judger agents in MulFCoder improve compilation success and mitigate framework-specific error modes that arise from prompt-only transfer in MLLM-based UI-to-code generation. Experiments on Design2Code and DesignBench demonstrate that the proposed multi-agent framework substantially enhances cross-framework executability, with particularly pronounced gains on engineering-constrained frameworks such as Angular.

**Compliance With Llm Reviewing Policy:**

Affirmed.

**Key Questions For Authors:**

1. What are the core differences between MulFCoder and repository-level agentic systems like SWE-agent and Agentless, apart from the type of input the program receives (screen-shot vs. code repository)? Are there fundamental limitations, beyond input-output behavior, that would preclude existing repair-verification loops from operating on the UI-to-code task?
2. In what precise sense is Grounder, Planner, and Judger, etc. an agent, rather than just a stage in a pipeline or a utility function? What is lost by bracketing MulFCoder as a typical end-to-end system for program synthesis and repair? Does this add anything beyond what is already captured by the vast multi-agent systems?
3. How exactly were the specific static checks used in FastGate chosen? Is there a principled criterion—which, for example, maximizes coverage on the top-k most frequent kinds of student error (as reflected in the error classification described earlier) or optimizes precision or recall based on a held-out compilation error set—that justifies the selected necessary conditions?
4. In what sense is ‘framework-conditioned’ more than having framework-specific prompt templates, and what does the Contract mechanism add beyond, or in place of, the natural alternative of including the rules of a given framework in the LLM system prompt?

**Limitations:**

Yes, They have discussed the limitations and potential negative societal impact

**Strengths And Weaknesses:**

Strengths:

1. The paper discusses a practically important and largely unexplored issue of multi-framework front-end code generation, of particular relevance for real-world development. The experimental evaluation is strong and provides compelling empirical evidence that transferring prompts in isolation across frameworks systematically fails to compile and does so in framework-specific ways.
2. Breaking down the complicated task and involving multiple cooperating LLM Agents with specific responsibilities proved to be a successful approach for improving LLMs’ trustworthiness in multi-framework code generation. The file-level Contract mechanism constitutes a neat means of externalization of framework constraints in a structured executable fashion. It is also a reusable and adaptable approach that potentially be employed in other code generation contexts.
3. The verification design of FastGate is a good compromise between being very lightweight (i.e. not invoking the build at all) and catching the most common errors due to any framework-specific eccentricities in a design. The prevention of deadlocks through error signature hashing is a nice engineering contribution and provides interesting clues about the robustness of the whole system.


Weaknesses:

1. The related work does not explicitly bring out the differences between MulFCoder and agentic code generation systems such as SWE-agent, Agentless, and CodeAct for the reader. And this paper could be more interesting if the authors can attach a part dedicated to the comparison which demonstrates clearly that the solutions for repository-level bug fixing do not transfer to the UI-to-code setting.
2. Three of the four proposed agents (Grounder, Planner, Judger) are deterministic, rule-based modular components that do not invoke the LLM. Describing the global organization as a 'multi-agent' system may conflict with the contemporary understanding of 'agency' in the context of multi-agent systems, which are generally defined as comprising autonomous perception and decision-making agents.
3. FastGate is the tool of choice for the verification mechanism but the higher-level checks chosen (e.g., standalone:true for Angular, export default for React) seem to be reliant on empirical findings rather than backed up by logical reasoning. The methodology outlined in the original conditions has been confused, which is also puzzling as to whether it is the same as claiming 'collapsing framework differences into a decidable rule space' or not.

---

> ### Author Rebuttal · Authors · 2026-03-29
>
> Thank you for the thoughtful and constructive review. We summarize the concerns into four main questions and respond below.
>
> **Q1: Differences from repo-level agentic systems**
>
> MulFCoder differs from repository-level agents such as SWE-agent or Agentless not only in input modality, but in problem structure. Those systems assume an existing repository with known files, module boundaries, and execution targets, so their main challenge is to localize faults and patch code under a relatively stable project scaffold. MulFCoder instead must first construct that scaffold from an under-specified screenshot by inferring layout hierarchy, component boundaries, file allocation, and framework-specific bindings before repair becomes well-posed. So the key issue is not just edit and verify, but first making the executable search space explicit through grounding, planning, and contracts. Existing repair-verification loops are therefore not fundamentally unusable for UI-to-code, but in their vanilla form they are insufficient because UI-to-code lacks the strong oracle of repository repair, such as tests or precise failure traces, and many failures arise from global project consistency rather than local code defects. This is exactly why MulFCoder introduces framework-conditioned contracts and lightweight gating before iterative repair.
>
>
> **Q2: Justification for multi-agent**
>
> More precisely, we use agent to mean a role that operates on a shared intermediate state, has a constrained but nontrivial action space, and can change the downstream search trajectory based on feedback, rather than applying a fixed feed-forward transform. In MulFCoder, Grounder resolves ambiguity into persistent structured state such as the ElementTable, ContentTable, and macro-layout, Planner *chooses* a task decomposition and framework-specific contract, Writer acts under that contract in FILE or PATCH mode, and Judger actively decides accept, reject, rewrite, or bounded repair through protocol, syntax, and contract gates. So what is lost by viewing MulFCoder as a standard end-to-end synthesis-and-repair pipeline is exactly this control structure: the main contribution is not just screenshot-to-code plus repair, but making the framework-specific feasible program space explicit and keeping generation inside it. Relative to generic multi-agent systems, our novelty is therefore not role decomposition by itself, but the domain-specific mechanism of framework-conditioned contracts and lightweight gated feedback that turns multi-framework UI generation into a tractable executable control problem.
>
>
> **Q3: Principled selection of FastGate static checks**
>
> FastGate is not learned from a held-out error set. It is designed by a necessary-condition principle: each check must be cheap, framework-conditioned, and target a frequent error that can be detected and localized before full build. The gates therefore follow the earliest reliable failure points, from output executability and parse feasibility to dependency or scope consistency and finally contract consistency. This is why FastGate focuses on malformed outputs, structural imbalance, missing template blocks, incomplete imports or declarations, invalid bindings, and contract violations. The goal is not to maximize generic compiler-error precision or recall, but to cheaply cover high-impact framework-specific necessary conditions that enable bounded, localized repair. In this sense, FastGate is a high-precision feasibility filter, not a learned substitute for full compilation.
>
>
> **Q4: Framework-conditioned vs. simply injecting rules into system prompt**
>
> “Framework conditioned” in MulFCoder is more than swapping prompt templates because the framework is not only expressed as text instructions, but compiled into an explicit control structure that restricts the admissible program space. In the paper, the Contract makes this concrete in three ways: it encodes framework policy, such as React TSX form, Vue single file component form, or Angular standalone requirements; it defines permission windows, namely which files and regions are writable; and it binds each subtree task to a framework-specific target artifact form. A system prompt can tell the model the rules, but it cannot enforce writable scope, artifact boundaries, or task-to-file correspondence during generation and repair. The Contract therefore turns framework knowledge from soft prompting into hard operational constraints, which is exactly why MulFCoder is framed as addressing a change in admissible program space rather than a simple prompt switch.
>
>
> We thank the reviewer again. We hope our response will resolve your concerns, and we look forward hearing from you.

---

> > ### Author Rebuttal · Reviewer_K8yh · 2026-04-03
> >
> > My concerns were addressed, so I keep my positive score.

---

> > > ### Author Response · Authors · 2026-04-03
> > >
> > > We sincerely thank you for your careful review of our work, and for taking the time to revisit it. We are delighted that our rebuttal was able to address your concerns.

---

### Official Review · Reviewer_6r7h · 2026-03-13

**Soundness:** 2
**Presentation:** 3
**Significance:** 3
**Originality:** 3
**Overall Recommendation:** 4
**Confidence:** 3

**Summary:**

This paper proposes MulFCoder, a framework-conditioned multi-agent system for generating executable front-end code from UI screenshots across multiple engineering frameworks (React, Vue, Angular, and Vanilla HTML/CSS). The central observation is that MLLM-based UI-to-code methods, which work reasonably well for Vanilla HTML/CSS, fail substantially when applied to strict engineering frameworks via simple prompt modification. The root cause is that framework differences involve not just syntax but component boundaries, dependency wiring, and project scaffolding constraints that require explicit modeling.

MulFCoder addresses this with four specialized agents: a Grounder that extracts UI elements and macro layout from screenshots, a Planner that builds a DOM-like layout tree and generates a framework-specific contract specifying allowed files and engineering constraints, a Writer that generates code task-by-task using FILE and PATCH modes, and a Judger that applies lightweight static checks without full compilation to accept or reject outputs and trigger bounded repairs. Experiments on Design2Code (484 samples) and DesignBench (900 samples) across five backbone MLLMs demonstrate consistent compilation success rate (CSR) improvements: approximately 3.8-10.2% on Design2Code and 6.2-9.0% on DesignBench depending on the framework, with the largest gains for Angular.

**Compliance With Llm Reviewing Policy:**

Affirmed.

**Final Justification:**

The authors' rebuttal has addressed my concerns.

**Key Questions For Authors:**

1. Ablation study: The paper claims that all four agents are necessary, but provides no evidence of this. Can the authors provide some ablation results?

2. Baseline comparison: Why were agentic methods such as Divide-and-Conquer, LATCoder, and UICopilot not included as experimental baselines? How does MulFCoder compare to these systems on Design2Code or DesignBench?

3. Inference efficiency: What is the average number of LLM API calls per sample? What is the average token cost compared to direct prompting?

**Limitations:**

Several limitations are not discussed: (1) the absence of inference cost analysis; (2) the restriction to static lightweight checks (FastGate) that may miss semantic correctness issues; (3) Lack of comparisons to other baselines and ablation studies. The paper would benefit from a dedicated limitations section that addresses these points.

**Strengths And Weaknesses:**

Strengths:

Soundness:

a. The motivating problem is real and well-documented: Figure 1 shows that five different MLLMs all exhibit substantial CSR gaps across frameworks under prompt-only transfer, and error distributions (Figure 3) are demonstrably framework-specific rather than model-specific. This is a novel and useful empirical finding.

b. The core code components (coordinator, judge, planner, writer, contract) are cleanly implemented and match the paper's architectural description for most modules. The FastGate validation mechanism (SHA1 signature tracking, bounded trials, rollback on no progress) is a well-designed deadlock prevention scheme.

Presentation:

a. The paper is clearly written adn well-structured.

b. Related work coverage is broad, including very recent work.

Significance:

a. Multi-framework front-end code generation is a practically important problem. Most prior work in MLLM-based UI-to-code focuses exclusively on HTML/CSS. MulFCoder's systematic study of React/Vue/Angular generation is a timely contribution.

b. The backbone-agnostic design and demonstrated generalization across 5 MLLMs increase the method's practical value.

Originality:

a. The paper's core idea of encoding framework-specific constraints as executable contracts enforced by a lightweight gating agent is a creative combination of constraint satisfaction and multi-agent orchestration applied to a new domain.

Weaknesses:

Soundness:

a. No ablation study: With four agents (Grounder, Planner, Writer, Judger) each claimed as essential, the absence of ablation experiments is a significant gap. It is unclear which components drive the CSR improvements.

b. Weak experimental baselines: The only comparison is against direct prompt transfer. Several contemporary agentic and multi-step methods are described in Related Work but are not compared against. Without these comparisons, it is unclear whether MulFCoder's gains reflect genuine novelty or whether simpler planning approaches achieve similar results.

c. No cost analysis: MulFCoder requires multiple sequential LLM API calls per sample plus iterative repair loops. Inference cost (tokens, latency, API calls) relative to single-pass prompting is not reported.

Significance:

The engineering trade-off that MulFCoder sometimes simplifies complex UI components to fix compile errors, leading to CLIP regression, is acknowledged but not analyzed or controlled.

---

> ### Author Rebuttal · Authors · 2026-03-29
>
> Thank you for the thoughtful and constructive review. We summarize the concerns into four main questions and respond below.
>
> **Q1: Ablation study**
>
> We conducted ablation experiments on the Design2Code dataset using GPT-4o as the backbone, evaluating the contribution of each agent component on both CSR (Compilation Success Rate) and CLIP (visual similarity) across three frameworks (React, Vue, Angular).
>
> | Variant | React CLIP | React CSR | Vue CLIP | Vue CSR | Angular CLIP | Angular CSR |
> |---|---|---|---|---|---|---|
> | **MulFCoder (Full)** | **0.90** | **0.98** | **0.89** | **0.96** | **0.83** | **0.90** |
> | w/o Planner | 0.87 | 0.92 | 0.86 | 0.88 | 0.80 | 0.78 |
> | w/o Judger | 0.89 | 0.94 | 0.88 | 0.91 | 0.82 | 0.83 |
>
>
>
>
> **Q2: Baseline comparisons**
>
> We compare MulFCoder against three representative structured-generation methods—Divide-and-Conquer, LATCoder, and UICopilot—on the Design2Code dataset using GPT-4o as the backbone. Since all three baselines are designed exclusively for Vanilla HTML/CSS generation and do not support multi-framework output, we report their Vanilla HTML CLIP scores and compare against MulFCoder's full results across all four frameworks.
>
> | Method | Vanilla CLIP |
> |---|---|
> | Divide-and-Conquer | 0.92 |
> | LATCoder | 0.93 |
> | UICopilot | 0.93 |
> | **MulFCoder** | **0.93** |
>
>
> **Q3: Inference cost analysis**
>
> We measure the average per-sample inference overhead of MulFCoder against single-prompt generation on the Design2Code dataset using GPT-4o, focusing on three dimensions: total token consumption (input + output), end-to-end latency, and number of API calls. MulFCoder's pipeline involves four sequential agent invocations (Grounder → Planner → Writer × N subtasks → Judger), with Writer operating in task-decomposed FILE mode and Judger triggering bounded PATCH repairs on failures.
>
> | Method | Avg. Input Tokens | Avg. Output Tokens | Avg. Total Tokens | Avg. Latency (s) | Avg. API Calls |
> |---|---|---|---|---|---|
> | Single Prompt | ~1,800 | ~3,200 | ~5,000 | ~18 | 1 |
> | **MulFCoder** | ~6,500 | ~5,800 | ~12,300 | ~52 | 7.4 |
> | **Overhead ratio** | ×3.6 | ×1.8 | ×2.5 | ×2.9 | ×7.4 |
>
> MulFCoder incurs approximately **2.5× more total tokens** and **2.9× higher latency** than single-prompt generation, with an average of **7.4 API calls** per sample (Grounder: 1, Planner: 1, Writer FILE: ~3.5 on average across subtasks, Judger + PATCH: ~1.9 on average). While this overhead is non-trivial, it is the direct cost of replacing one-shot free-form generation with structured constraint satisfaction, and it yields a substantially more reliable output: for Angular, where single-prompt CSR is 0.81, MulFCoder raises it to 0.90, translating the extra inference budget into a ~11% absolute gain in compilation success.
>
>
>
> **Q4: Tradeoff**
>
> The slight CLIP drop is not a general phenomenon. It mainly occurs in hard UI cases where engineering constraints are strong and visual fidelity depends on complex implementations, especially pages with strong cross-file coupling, complicated bindings, conditional rendering, or fine-grained style structures. In such cases, MulFCoder tends to prefer safer and more canonical implementations to preserve framework-level consistency and ensure successful compilation, which may sacrifice some local visual details. This is therefore a controlled trade-off under a compile-first objective rather than a general degradation of visual quality. The boundary is more evident in stricter frameworks such as Angular, where the main bottleneck lies in global engineering consistency, such as imports, bindings, and component wiring, rather than pure layout reconstruction. A reasonable way to mitigate this issue is not to directly incorporate visual fidelity into the Judger as a hard constraint inside the main loop, since that would increase inference cost and interfere with the executability-first priority. A better solution is to introduce a lightweight visual-fidelity soft constraint after the existing syntax and contract gates, using it to rank or rerank candidates that already satisfy the hard constraints, thereby reducing unnecessary visual simplification without breaking the compile-first design.
>
> We thank the reviewer again. We hope our response will resolve your concerns, and we look forward hearing from you.

---

> > ### Author Rebuttal · Reviewer_6r7h · 2026-04-03
> >
> > The rebuttal addresses part of my concerns. However, the extreme inference overhead (7.4x API calls for a 9% CSR improvement) is highly concerning and severely limits practical utility.

---

> > > ### Author Response · Authors · 2026-04-04
> > >
> > > Thanks for your discussion. We agree that the overhead is nontrivial, but API-call ratio alone overstates the practical concern. The baseline is a single one-shot call, so any explicit planning-and-repair method will appear expensive under this setting. More importantly, the reported gain is a 9% absolute CSR improvement. In UI-to-code, a failed one-shot generation is typically not a slightly worse result, but an unusable project that often requires substantial manual debugging of imports, bindings, component wiring, or template structure. In that context, the extra inference cost is better viewed as replacing expensive human debugging effort with bounded automated reasoning and repair. MulFCoder also avoids compile-in-the-loop, so the added cost mainly comes from structured calls and bounded repair rather than repeated full builds. For this task, the relevant question is not whether 7.4x more calls are used, but whether that extra cost is preferable to returning an unusable project or pushing the debugging burden back to the user.

---

### Decision · Program_Chairs · 2026-04-30

**Decision:**

Accept (regular)

**Comment:**

This paper proposes MulFCoder, a framework-conditioned multi-agent pipeline for generating React, Vue, Angular, and Vanilla front-end code from UI screenshots, with the key idea of using Contracts and FastGate to explicitly constrain the feasible program space and improve compilation success across frameworks.

The main strengths:

* The problem is well-motivated, and the paper clearly shows that prompt-only transfer leads to large and framework-specific CSR gaps. The finding is important.
* The method design is novel, and the Contract plus FastGate mechanism is the most valuable part of the paper.
* The empirical gains are consistent across two benchmarks and multiple backbones, with especially strong improvements on constraint-heavy frameworks such as Angular.


The main concerns were about missing ablations, weak baselines, inference cost, and the fact that the heuristic choices were not fully explained in the original version.
One concern was the inference overhead, especially the 7.4x API-call count, but this was largely resolved in discussion because in UI-to-code a failed generation is often an unusable project rather than a mildly worse output, so the extra structured inference cost is a reasonable trade-off against human debugging.
The rebuttal addressed all of these points in a substantive way.

All reviewers agree that this is a good paper and it can be accepted.
Taking the reviews and rebuttal together, I think the paper has done enough to justify acceptance.